# Simplifying Automated Pattern Selection for Planning with Symbolic Pattern Databases

**Ionut Moraru**[1], **Stefan Edelkamp**[1], **Moises Martinez**[1] and **Santiago Franco**[2]

[1]Informatics Department, Kings College London, UK
[2]School of Computing and Engineering, University of Huddersfield, UK
{firstname.lastname}@kcl.ac.uk, s.franco@hud.ac.uk

## Abstract

Pattern databases (PDBs) are memory-based abstraction heuristics that are constructed prior to the planning process which, if expressed symbolically, yield a very efficient representation. Recent work in the automatic generation of symbolic PDBs has established it as one of the most successful approaches for cost-optimal domain-independent planning. In this paper, we contribute two planners, both using *bin-packing* for its pattern selection. In the second one, we introduce a greedy selection algorithm called *Partial-Gamer*, which complements the heuristic given by bin-packing. We tested our approaches on the benchmarks of the last three International Planning Competitions, optimal track, getting very competitive results, with this simple and deterministic algorithm.

## 1 Introduction

The automated generation of search heuristics is one of the holy grails in AI, and goes back to early work of Gaschnik (1979), Pearl (1984), and Prieditis (1993). In most cases lower bound heuristics are problem relaxations: each plan in the original state space maps to a shorter one in some corresponding abstract one. In the worst case, searching the abstract state spaces at every given search nodes exceeds the time of blindly searching the concrete search space (Valtorta 1984). With pattern databases (PDBs), all efforts in searching the abstract state space are spent prior to the plan search, so that these computations amortize through multiple lookups.

Initial results of Culberson and Schaeffer (1998) in sliding-tile puzzles, where the concept of a pattern is a selection of tiles, quickly carried over to a number of combinatorial search domains, and helped to optimally solve random instances of the Rubik's cube, with non-pattern labels being removed (Korf 1997). When shifting from breadth-first to shortest-path search, the exploration of the abstract state-space can be extended to include action costs.

The combination of several databases into one, however, is tricky (Haslum et al. 2007). While the maximum of two PDBs always yields a lower bound, the sum usually does not. Korf and Felner (2002) showed that with a certain selection of disjoint (or additive) patterns, the values in different PDBs can be added while preserving admissibility. Holte et al. (2004) indicated that several smaller PDBs may out-perform one large PDB. The notion of a pattern has been generalized to production systems in vector notation (Holte and Hernádvölgyi 1999), while the automated pattern selection process for the construction of PDBs goes back to the work of Edelkamp (2006).

Many planning problems can be translated into state spaces of finite domain variables (Helmert 2004), where a selection of variables (pattern) influences both states and operators. For disjoint patterns, an operator must distribute its original cost, if present in several abstractions (Katz and Domshlak 2008; Yang et al. 2008).

During the PDB construction process, the memory demands of the abstract state space sizes may exceed the available resources. To handle large memory requirements, symbolic PDBs succinctly represent state sets as binary decision diagrams (Edelkamp 2002). However, there are an exponential number of patterns, not counting alternative abstraction and cost partitioning methods. Hence, the automated construction of informative PDB heuristics remains a combinatorial challenge. Hill-climbing strategies have been proposed (Haslum et al. 2007), as well as more general optimization schemes such as genetic algorithms (Edelkamp 2006; Franco et al. 2017). The biggest issue in this area remains assessing the quality of the PDBs (in terms of the heuristic values for the concrete state space) which can only be estimated. Usually, this involves generating the PDBs and evaluating them (Edelkamp 2014; Korf 1997).

This work contributes by improving the automated pattern selection process. We first define the settings of cost-optimal action planning and give a characterization of a pattern database. We stress spurious states, as they are inevitable to PDB generation. Next, we move to the encoding of the pattern selection problem and how to evaluate the heuristics resulted from them. The main contribution is a greedy partial PDB selection mechanism, which we show that complements well with bin packing, giving close to state of the art results on our benchmarks (bettering the results of the winner of the 2018 International Planning Competition).

## 2 Background

There are a variety of planing formalisms. Fikes and Nilson (1971) invented the propositional specification language STRIPS, inspiring PDDL (McDermott 1998). Holte and

Hernádvölgyi (1999) invented the production system vector notation (PSVN) for permutation games. Bäckström (1992) prefers the SAS$^+$ formalism, which is a notation of finite-domain state variables over partial states and operators with pre-, (prevail-,) and postconditions.

**Definition 1 (SAS$^+$ Planning Task)** *is a quadruple* $\mathcal{P} = \langle \mathcal{V}, \mathcal{O}, s_0, s_* \rangle$, *where* $\mathcal{V} = \{v_1, \ldots, v_n\}$ *is the set of finite-domain variable;* $\mathcal{O}$ *are the operators which consist of pre-conditions and effects. The remaining two,* $s_0$ *and* $s_*$ *are states. A (complete) state* $s = (a_1, \ldots, a_n) \in \mathcal{S}$ *assigns a value* $a_i$ *to every* $v_i \in \mathcal{V}$, *with* $a_i$ *in a finite domain* $D_i$, $i = 1, \ldots, n$. *For partial states* $s^+ \in \mathcal{S}^+$, *each* $v_i \in \mathcal{V}$ *is given an extended domain* $D_i^+ = D_i \cup \{\_\}$. *We have* $s_0 \in \mathcal{S}$ *and* $s_* \in \mathcal{S}^+$.

*A* state space abstraction $\phi$ *is a mapping from states in the original state space* $\mathcal{S}$ *to the states in the abstract state space* $\mathcal{A}$.

*Let an abstract operator* $o' = \phi(o)$ *be defined as* $pre' = \phi(pre)$, *and* $post' = \phi(post)$. *For planning task described above, the corresponding abstract task is* $\langle \mathcal{V}, \mathcal{O}', s_0', s_*' \rangle$ *with* $s_0' \in \mathcal{A}$, $s_*' \in \mathcal{A}^+$, *The result of applying operator* $o' = (pre', post')$ *to an abstract state* $a = s'$ *satisfying* $pre'$, *sets* $s_i' = post_i' \neq \_$, *for all* $i = 1, \ldots, n$.

A cost is assigned to each operator. In the context of cost-optimal planning, the aim is to minimize the total cost over all plans that lead from the initial state to one of the goals.

The set of reachable states is generated on-the-fly, starting with the initial state by applying the operators. In most state-of-the-art planners, lifted planning tasks are grounded to SAS$^+$. A STRIPS domain with states being subsets of propositional atoms can be seen as a SAS$^+$ instance with a vector of Boolean variables. The core aspect of grounding is to establish invariances, which minimizes the SAS$^+$ encoding.

**Definition 2 (State-Space Homomorphism)** *A homomorphic abstraction* $\phi$ *imposes that if* $s'$ *is the successor of* $s$ *in the concrete state space we have* $\phi(s')$ *is the successor of* $\phi(s)$ *in abstract one. This suggests abstract operators* $\phi(o)$ *leading from* $\phi(s)$ *to* $\phi(s')$ *for each* $o \in \mathcal{O}$ *from* $s$ *of* $s'$.

As the planning problem spans a graph by applying a selection of set of rules, the planning task abstraction is generated by abstracting the initial state, the partial goal state *and* the operators. Plans in the original space have counterparts in the abstract space, but not vice verse. Usually, the planning task of finding a plan from $\phi(s_0)$ to $\phi(s_*)$ in $\mathcal{A}$ is computationally easier than finding one from $s_0$ to $s_*$ in $\mathcal{P}$.

The main issue encountered when working with abstractions are *spurious* paths in the abstract state space that have no corresponding path in the original (concrete) space. An intuitive example of two disconnected paths $s_0 \rightarrow s_1 \rightarrow s_2 \rightarrow s_3 \rightarrow \ldots \rightarrow s_l$ and $s_{l+1} \rightarrow s_{l+2} \rightarrow s_{l+3} \rightarrow \ldots \rightarrow s_m = s_*$, is shown in Figure 1 with $l = 3$. As we map $s_l$ and $s_{l+1}$ to the same abstract state, we have an abstract plan which has no preimage in the original one.

Homomorphic abstractions preserve the property that every path (plan) present in the original space is also present (and shorter) in the abstract state space. Still, abstract operators may yield spurious states.

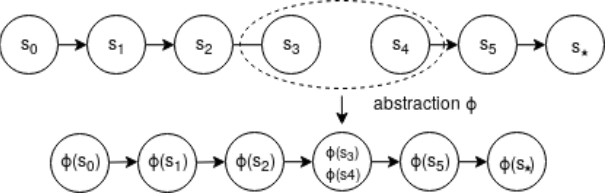

Figure 1: Example of the spurious path problem.

This problem also pops up in unabstracted search spaces. One cause for this is the nature of PDB construction, namely regression search. To illustrate this, consider the $(1 \times 3)$ sliding-tile puzzle with two tiles 1 and 2 and one empty position, the blank. In one SAS$^+$ encoding we have three state variables: two for the position of the tile $t_i \in \{1, 2, 3\}$, $i \in \{1, 2\}$, and one for the position of the blank $b \in \{1, 2, 3\}$. Let $s_0 = (t_1, t_2, b) = (2, 3, 1)$ and $s_* = (1, 2, \_)$. The operators have preconditions $t_i = x$, $b = x + 1$, and effects $t_i = x + 1$, $b = x$, or preconditions $t_i = x$, $b = x - 1$, and effects $t_i = x - 1$, $b = x$, for $i = \{1, 2\}$ and $x \in \{1, 2, 3\}$ (whenever possible). Going backwards from $s_*$, the planner does not know the location of the blank and beside the reachable state $t_1 = 2$, $t_2 = 3$, $b = 3$ it generates two additional states $t_1 = t_2 = 1$, $b = 2$ and $t_1 = t_2 = 2$, $b = 1$.

How to mitigate the problem? We cannot expect to remove all spurious states, but there is hope to reduce their number. In the case of the sliding-tile puzzle, there is a dual SAS$^+$ encoding with three variables denoting which tile (or blank) is present at a given position $p_1$, $p_2$, or $p_3$. This *exactly-one-of* state invariance is inferred by the static analyzer, but not used in the state encoding. The information, however, can help to eliminate spurious states.

Either spurious paths through abstraction or though regression, they do not affect the lower bound property of the resulting abstraction heuristic. However, they can blow up the PDBs considerably, given that there are abstract states and paths for which no corresponding preimage in the forward space exist. As a result, refined state invariants (including mutex detection of contradicting facts) greatly improve backward search and, thus, reduce the size of pattern databases.

## 3 Pattern Databases

As planning is a PSPACE-complete problem (Bylander 1994), heuristic search has proven to be one of the best ways to find solutions in a timely manner.

**Definition 3 (Heuristic)** *A heuristic* $h$ *is a mapping of the set of states in* $\mathcal{P}$ *to positive reals* $R_{\geq 0}$. *A heuristic is called* admissible, *if* $h(s)$ *is a lower bound of the cost of all goal-reaching plans starting at* $s$. *Two heuristics* $h_1$ *and* $h_2$ *are* additive, *if* $h$ *defined by* $h(s) = h_1(s) + h_2(s)$ *for all* $s \in \mathcal{S}$, *is admissible. A heuristic is* consistent *if for all operators* $o$ *from* $s$ *to* $s'$ *we have* $h(s') - h(s) + c(o) \geq 0$.

For admissible heuristics, search algorithms like A* (Hart, Nilsson, and Raphael 1968) will return optimal plans. If $h$ is also consistent, no states will be reopened during search. This is the usual case for PDBs.

**Definition 4 (Pattern Database)** *is an abstraction mapping for states and operators* and *a lookup table that for each abstract state $a$ provides the (minimal) cost value from $a$ to the goal state.*

The minimal cost value is a lower bound for reaching the goal of the state that is mapped to $a$ in the original state space. PDBs are generated in a backwards enumeration of the abstract state space, starting with the abstract goal. They are stored in a (perfect) hash table for explicit search, and in the form of a BDD with all abstract states of a certain $h$ value while in symbolic search.

Showing that PDBs yield consistent heuristics is trivial (Edelkamp 2014; Haslum et al. 2005), as shortest path distances satisfy the triangular inequality. It has also been shown that for PDBs the sum of heuristic values obtained via *projection* to a disjoint variable set is admissible (Edelkamp 2014). The projection of state variables induces a projection of operators and requires *cost partitioning*, which distributes the cost $c(o)$ of operators $o$ to the abstract state spaces (Pommerening et al. 2015). We will discuss more about cost partitioning in section 4.

For ease of notation, we identify a pattern database with its abstraction function $\phi$. As we want to optimize PDBs via genetic algorithms, we need an objective function.

**Definition 5 (Average Fitness of PDB)** *The* average fitness $f_a$ *of a PDB $\phi$ (interpreted as a set of pairs $(a, h(a))$) is the average heuristic estimate $f_a(\phi) = \sum_{(a,h(a))\in\phi} h(a)/|\phi|$, where $|\phi|$ denotes the size of the PDB $\phi$.*

There is also the option of evaluating the quality of PDB based on a sample of paths in the original search space.

**Definition 6 (Sample Fitness of PDB)** *The fitness $f_s$ of a PDB $\phi$ wrt. a given sample of (random) paths $\pi_1, \ldots, \pi_m$ and a given candidate pattern selection $\phi_1, \ldots, \phi_k$ in the search space is determined by whether the number of states with a higher heuristic value (compared to heuristic values in the existing collection) exceeds a certain threshold $C$, i.e.,*

$$\sum_{i=1}^{m}[h_\phi(last(\pi_i)) > \max_{j=1}^{k}\{h_{\phi_j}(last(\pi_i))\}] > C,$$

*where $[cond] = 1$, if cond is true, otherwise $[cond] = 0$, and $last(\pi)$ denotes the last state on $\pi$.*

**Definition 7 (Pattern Selection Problem)** *is to find a collection of PDBs that fit into main memory, and maximize the average heuristic value[1].*

**Definition 8 (Perimeter PDB)** *is the result of an unabstracted (blind) backward shortest path search until memory resources are exhausted, setting the value of all yet unreached abstract space to the maximum cost value found in the perimeter, while adding the minimum cost of an operator.*

In several planning tasks, generating the perimeter PDB already solved the problem (Franco et al. 2017).

---

[1]The average heuristic value has shown empirically that it is a good metric. While it is not the solution to evaluating the pattern selection problem perfectly, it is a good approximation up to this point.

## Symbolic Pattern Databases

In symbolic plan search, we encode each variable domain $D_j$ of the SAS$^+$ encoding, $j = 1, \ldots, n$, in binary. Then we assign a Boolean variable $x_i$ to each $i$, $0 \leq i < \lceil\log_2|D_1|\rceil + \ldots + \lceil\log_2|D_n|\rceil$. This eventually results in a characteristic function $\chi_S(x)$ for any set of states $S$. The ordering of the variables is important for a concise representation, so that we keep finite domain variables as blocks and move inter-depending variables together. The optimization problem of finding such best linear variable arrangement among them is NP-hard. It is also possible to encode operators as Boolean functions $\chi_o(x, x')$ and to progress (and regress) a set of states to accelerate this (pre)image, the disjunction of the individual operators images could be optimized. For action costs, always expanding the set attached to the minimum cost value yields optimal results (Edelkamp 2002). As symbolic search is available for partial states (which denote sets of states), both the forward and the backward symbolic exploration in plan space become similar.

There has been considerable effort to show that PDB heuristics can be generated symbolically and used in a symbolic version of A* (Edelkamp 2002). The concise representation of the Boolean formula for these characteristic functions in a binary decision diagram (BDD) is a technique to reduce the memory requirement during the search. Frequently, the running time for the exploration often reduces as well.

## 4 Pattern Selection and Cost Partitioning

Using multiple abstraction heuristics can lead to solving more complex problems, but to maintain optimality, we need to distribute the cost of an operator among the abstractions. One way of doing this is present in (Seipp and Helmert 2018). Saturated Cost Partitioning (SCP) has shown benefits to simpler cost partitioning methods. Given an ordered set of heuristics, in our case PDBs, SCP relies on only using those costs which each heuristic uses to create an abstract plan. The remaining costs are left free to be used by any subsequent heuristic. However, considering the limited time budget, this approach is more time consuming compared to other cost partitioning methods (Seipp, Keller, and Helmert 2017).

One such method is 0/1 cost partition, which zeroes any cost for subsequent heuristics if the previous heuristic has any variables affected by that operator. Both SCP and 0/1 allow heuristics values to be added admissibly. SCP dominates 0/1 cost partitioning (given a set of patterns and enough time, SCP would produce better heuristic values), but it is much more computationally expensive than 0/1 cost partitioning.

Franco et al., (2017) shows that, in order to find good complementary patterns, it is beneficial to try as many pattern collections as possible. As such, we implemented 0/1 cost partitioning in our work. We tested using the canonical cost partitioning (Haslum et al. 2007) method as well whenever we added a new PDB, but this resulted in a very pronounced slow down which increased the more PDBs have

already been selected. This was the reason we adopted a hybrid combination approach, where 0/1 cost partition is used on-the-fly to generate new pattern collections, and, only after all interesting pattern collections have been selected, we run the canonical combination method, slightly extended to take into account that each pattern has its own 0/1 cost partition.

Given a number of PDBs in the form of pattern collections (sets of individual patterns, each associated with a cost partitioning function), *canonical pattern databases* will select the best admissible combination of PDB maximization and addition. The computation of the canonical PDB is still expensive, so we execute it only once, right before search starts.

There are many alternatives for automated pattern selection based on bin packing such as random bin packing (PBP), causual dependency bin packing (CBP), which could be refined by a genetic algorithm (Franco et al. 2017).

### Greedy Selection

Franco et al. (2017) compared the pattern selection method to the one of Gamer (Kissmann and Edelkamp 2011), which tries to construct one single best PDB for a problem. Its pattern selection method is an iterative process, starting with all the goal variables in one pattern, where the causally connected variables who would most increase the average $h$ value of the associated PDB are added to the pattern.

Following this work, we devised a new *Gamer-style* pattern generation method, which behaves similarly, but which adds the option of *partial pattern database* generation to it. By partial we mean that we have a time and memory limit for building each PDB. If the PDB building goes past this limit, we truncate it in the same way we would do with a perimeter PDB, i.e., any unmapped real state has the biggest $h$ value the PDB building was at when it was interrupted.

An important difference with the Gamer method is that we do not try every possible pattern resulting of adding a single causally connected variable to the latest pattern.

### Genetic Algorithm Selection

A *genetic algorithm* (GA) is a general optimization method in the class of *evolutionary strategies* (Holland 1975). It refers to the recombination, selection, and mutation of *genes* (states in a state-space) to optimize the *fitness* (objective) function. In a GA, a population of candidate solutions is sequentially evolved to generate a better performing population of solutions, by mimicking the process of evolution. Each candidate solution has a set of properties which can be mutated and recombined. Traditionally, candidate solutions are bitvectors, but there are strategies that work on real-valued state vectors.

An early approach for the automated selection of PDB variables by Edelkamp (2006) employed a GA with genes representing state-space variable patterns in the form of a 0/1 matrix $G$, where $G_{i,j}$ denotes that state variable $i$ is chosen in PDB $j$. Besides flipping and setting bits, mutations may also add and delete PDBs in the set.

The PDBs corresponding to the bitvectors in the GA have to fit into main memory, so we have to restrict the generation

---

**Algorithm 1** Greedy PDBs Creation

1: **function** GREEDYPDBS($M$,$T$,$S_{min}$,$S_{max}$,$EM$) :
**Require:** time and memory limits $T$ and $M$, min and max PDB size $S_{min}$ ad $S_{max}$, evaluation method $EM$.
2:  $SelPDBs \leftarrow \emptyset$
3:  $\mathcal{P}_{sel} \leftarrow \mathcal{P}_{sel} \cup Packer(FFD, S_{min}, M, T, EM)$
4:  $\mathcal{P}_{sel} \leftarrow \mathcal{P}_{sel} \cup Packer(FFI, S_{min}, M, T, EM)$
5:  $\mathcal{P}_{sel} \leftarrow \mathcal{P}_{sel} \cup PartialGamer(M, T, EM)$
6:  **Return** $\mathcal{P}_{sel}$
7: **end function**

8:
9: **function** PACKER($Method$,$S_{min}$, $M$, $T$,$EM$) :
10:  $SizeLim \leftarrow S_{min}$
11:  **while** $(t < T)$ **and** $(m < M)$ **do**
12:    GENERATE_$\mathcal{P}$($Method$,$SizeLim$)
13:    **if** $EM(\mathcal{P})$ **then**
14:      $\mathcal{P}_{sel} \leftarrow \mathcal{P}$
15:    **end if**
16:    $Size \leftarrow Size * 10$
17:  **end while**
18:  **Return** $\mathcal{P}_{sel}$
19: **end function**

20:
21: **function** PARTIALGAMER($M$, $T$,$EvalMethod$) :
22:  $InitialPDB \leftarrow$ all goal variables
23:  $SelPDBs \leftarrow InitialPDB$
24:  **while** $(t < T)$ **and** $(m < M)$ **do**
25:    generate all $CandidatePatterns$ resulting of adding one casually connected variable to latest $P \in \mathcal{P}_{sel}$
26:    **for all** $P \in CandidatePatterns$ **do**
27:      **if** $EM(\mathcal{P})$ **then**
28:        $\mathcal{P}_{sel} \leftarrow P$
29:        break
30:      **end if**
31:    **end for**
32:  **end while**
33:  **Return** $\mathcal{P}_{sel}$
34: **end function**

---

of offsprings to the ones that represent a set of PDB that respect the memory limitation. If time becomes an issue, we stop evolving patterns and invoke the overall search (in our case progressing explicit states) eventually. An alternative, which sometimes is applied as a subroutine to generate the initial population for the GA, is to use bin packing.

### Bin Packing

The bin packing problem (BPP) is one of the first problems shown to be NP-hard (Garey and Johnson 1979). Given objects of integer size $a_1, \ldots, a_n$ and maximum bin size $C$, the problem is to find the minimum number of bins $k$ so that the established mapping $f : \{1, \ldots, n\} \rightarrow \{1, \ldots, k\}$ of objects to bins maintains $\sum_{f(a)=i} a \leq C$ for all $i \leq k$. The problem is NP-hard in general, but there are good approximation strategies such as first-fit and best-fit decreasing (being at most 11/9 off the optimal solution (Dósa 2007)).

In the PDBs selection process, however, the definition of the BPP is slightly different. We estimate the size of the PDB by computing the product (not the sum) of the variable domain sizes, aiming for a maximum bin capacity $M$ imposed by the available memory, and we find the minimum number of bins $k$, so that the established mapping $f$ of objects to bins maintains $\prod_{f(a)=i} a \leq M$ for all $i \leq k$. By taking the logs on both sides, we are back to sums, but the sizes become fractional. In this case, $\prod_{f(a)=i}$ is an upper bound on the number of abstract states needed.

Taking the product of variable domain sizes is a coarse upper bound. In some domains, the abstract state spaces are much smaller. Bin packing chooses the memory bound on each individual PDB, instead of limiting their sum. Moreover, for symbolic search, the correlation between the cross product of the domains and the memory needs is rather weak. However, because of its simplicity and effectiveness, this form of bin packing currently is chosen for PDB construction.

By limiting the amount of optimization time for each BPP, we do not insist on optimal solutions, but we want fast approximations that are close-to-optimal. Recall, that suboptimal solutions to the BPP do not imply suboptimal solutions to the planning problem. In fact, *all* solutions to the BPP lead to admissible heuristics and therefore optimal plans.

For the sake of generality, we strive for solutions to the problem that do not include problem-specific knowledge but still work efficiently. Using a general framework also enables us to participate in future solver developments. Therefore, in both of the approaches we present in this paper, we focus on the first-fit algorithm.

First-Fit Increasing (FFI), or Decreasing (FFD), is a fast on-line approximation algorithm that first sorts the objects according to their sizes and, then, starts placing the objects into the bins, putting an object to the first bin it fits into. In terms of planning, the variables are sorted by the size of their domains in increasing/decreasing order. Next, the *first* variable is chosen and packed at the same bin with the rest of the variables which are related to it if there is space enough in the bin. This process is repeated until all variables are processed.

## 5  Symbolic PDB Planners

Based on the results from (Franco et al. 2017), we decided to work only with Symbolic PDBs. Further experiments suggested that PDBs heuristic performs well when it is complemented with other methods. One good combination was using our method to complement a symbolic perimeter PDB, method that we used in the first of the planners we present. The selected method to be complemented first generates a symbolic PDB up to a fixed time limit and memory limit. One advantage of seeding our algorithm with such a perimeter search is that if there is an easy solution to be found in what is basically a brute force backwards search, we are finished before even creating a PDB. Secondly, we combined the Partial-Gamer with bin packing and saw very good results in how they complemented each other. In Figure 2 we see that each method gives good results on their own, Bin-

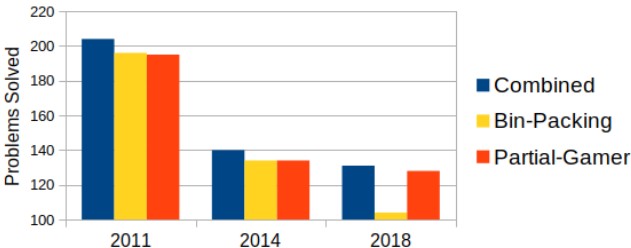

Figure 2: Coverage of Bin Packing, Partial Gamer and of both combined on three latest cost-optimal IPC benchmark problems.

Packing solving 434 and Partial-Gamer 457, but when used together they increase to 475.

In our work, however, we decided to use a hybrid, keeping the forward exploration explicit-state, and the PDBs generated in the backward exploration symbolic. Lookups are slightly slower than in hash tables, but they are still in time linear to the bitvector length.

In this section, we will present two symbolic planners, Planning-PDBs and GreedyPDB, based on the Fast-Downward planning framework (Helmert 2006). The two differ in the pattern selection methods that we use in each of them.

### GreedyPDB

We encountered that greedily constructed PDBs outperform the perimeter PDB, which we decided not to use. The two construction methods do not complement well, on the extreme case greedy PDBs will build a perimeter PDB after adding all the variables. There is a significant amount of overlapping between both methods. The collection of patterns received from bin packing, however, complements well the greedily constructed PDBs. One reason for this is that in domains amenable to cost-partitioning strategies, i.e. alternative goals are easily parallelized into a complementary collection of PDBs, bin packing can do significantly better than the single PDB approach. Evaluation is based on the definition of sample fitness. The sample is redrawn each time an improvement was found.

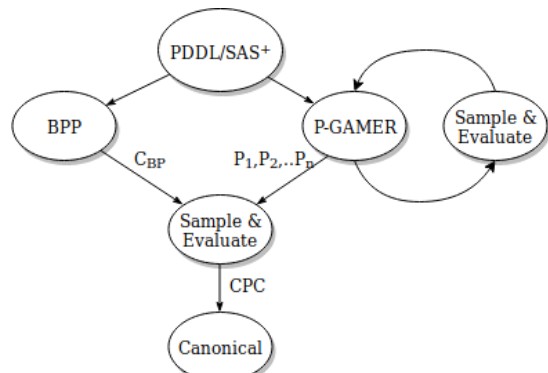

Figure 3: High level architecture of GreedyPDB

Algorithm 1 shows how Greedy PDBs combines two bin packing algorithms with a greedy selection method called

| Domain/Method | Agr | Cal | DN | Nur | OSS | PNA | Set | Sna | Spi | Ter | Total |
|---|---|---|---|---|---|---|---|---|---|---|---|
| GreedyPDB | 13 | **12** | **14** | **15** | 13 | 16 | 8 | 13 | 11 | 16 | **131** |
| BP-PDB | 6 | 12 | 14 | 12 | **13** | 19 | 8 | 11 | 12 | 16 | 123 |
| Scorpion | 1 | **12** | **14** | 12 | 13 | 0 | **10** | 13 | **15** | 14 | 104 |
| SymBiDir | **14** | 9 | 12 | 11 | **13** | 19 | 8 | 4 | 6 | **18** | 114 |
| Complementary1 | 10 | 11 | 14 | 12 | 12 | 18 | 8 | 11 | 11 | 16 | 123 |
| Complementary2 | 6 | **12** | 13 | 12 | **13** | 18 | 8 | **14** | 12 | 16 | 124 |
| Oracle | 14 | 12 | 14 | 15 | 13 | 19 | 10 | 14 | 15 | 18 | 142 |

Table 1: Coverage of PDB-type planners on the 2018 International Planning Competition for cost-optimal planning

Partial Gamer. The two bin packing algorithms use First Fit Decreasing (*FFD*) and First Fit Increasing (*FFI*), same used in Planning-PDBs. For FFD we set a limit of 50 seconds, while for FFI we used a limit of 75 seconds (both limits were found empirically to give the best results). To evaluate (*EM*) if the generated pattern collections should be added to our selection ($\mathcal{P}_{sel}$), we used as an evaluation method a random walk. If enough of the sampled states heuristic values are improved, the pattern is selected. Partial Gamer greedily grows the largest possible PDB by adding causally connected variables to the latest added pattern. If a pattern is found to improve, as defined by the evaluation method, then we add it to the list of selected pattern collections as a pattern collection with a single PDB. Note that we are using symbolic PDBs with time limits on PDB construction, hence a PDB which includes all variables of a smaller PDB does not necesarily dominate it since the smaller PDB might reach a further depth.

An important difference with the Gamer method is that we do not try every possible pattern resulting of adding a single causally connected variable to the latest pattern. As soon as a variable is shown to improve the pattern, we add it and restart the search for an even larger improving pattern. We found this to work better with the tight time limits required by combining several approaches. All the resulting pattern database collections are combined by simply maximizing their individual heuristic values. The PDBs inside each collection were combined using zero-one cost partitioning. The rationale behing the algorithm is that some domains are more amenable to using several patterns where costs are distributed between each patterns, while other domains seem to favour looking for the best possible single pattern.

### Planning-PDBs

In *Planning-PDBs*[2], we start with the construction of the perimeter PDB, and continue by using two bin-packing methods to create a collection of PDBs. The first method uses first-fit increasing, while the second being first-fit decreasing. Bin-packing for PDBs creates a small number of PDBs which use all available variables. Even though reducing the number of PDBs used to group all possible variables does not guarantee a better PDB, by having a smaller PDB collections, it is less likely to miss interactions between variables due to them being placed on different PDBs. The bin

packing algorithms used ensures that each PDB has a least one goal variable.

If no solution is found after the perimeter PDB has been finished, the method will start generating pattern collections stochastically until either the generation time limit or the overall PDB memory limit are reached. We then decide whether to add a pattern collection to the list of selected patterns if it is estimated that adding such PDB will speed up search. We optimize the results given by the bin-packing algorithm giving it to a GA. It then resolve operator overlaps in a 0/1 cost partitioning. To evaluate the fitness function, the corresponding PDBs is built —a time-consuming operation, which nevertheless payed off in most cases. Once all patterns have been selected, the resulting canonical PDB combination is used as an admissible heuristic to do A* search.

## 6   Experiments

Following is an ablation-type study were we analyze which components worked best. We run different configurations on the competition benchmarks on our cluster that utilized Intel Xeon E5-2660 V4 with 2.00GHz processors. We compare GreedyPDB and Planning-PDBs with other pattern database and symbolic planners that competed in the 2018 International Planning Competition in the most prestigious and attended deterministic cost-optimal track.

| Year/Method | 98-09 | 2011 | 2014 | 2018 | Total |
|---|---|---|---|---|---|
| GreedyPDB | 665 | **204** | 140 | **131** | 1140 |
| Planning-PDB | 678 | 190 | 131 | 123 | 1122 |
| Scorpion | **785** | 190 | 118 | 104 | **1197** |
| SymBiDir | 686 | 174 | 129 | 114 | 1064 |
| Comp1 | 680 | 185 | 111 | 123 | 1099 |
| Comp2 | 686 | **204** | **155** | 124 | 1169 |
| Oracle | 820 | 227 | 171 | 143 | 1361 |

Table 2: Overall coverage of PDB-type planners across different International Planning Competitions for cost-optimal planning. All benchmark sets are complete except for the 98-09, in which we use 31 of the domains

Looking at the results of various cost-optimal planners across all domains from the IPC competitions from 1998 to 2018 in Table **??**, we get a good overall picture on the PDB planner performance. Symbolic bidirectional, the benchmark planner in the IPC18 (1064 problems solved overall, 412 for the last 3 IPCs) is almost on par with Scorpion (412) and Complementary1 (419) on the last 3 IPCs, but when adding the 98-09 domains it falls last compared with all

---

[2]This planner has competed in the 2018 IPC on the Optimal track (Martinez et al. 2018) - https://tinyurl.com/PlanningPDBs

| Planner/Domain | Greedy PDB | Greedy PDB BinPack | Greedy PDB PartGamer | Planning PDB | Scorpion | SymBiDir | Comp1 | Comp2 | Oracle |
|---|---|---|---|---|---|---|---|---|---|
| Agr | 13 | 4 | 8 | 6 | 1 | **14** | 10 | 6 | 14 |
| Cal | **12** | **12** | **12** | **12** | **12** | 9 | 11 | **12** | 12 |
| DN | **14** | **14** | 12 | **14** | **14** | 12 | **14** | 13 | 14 |
| Nur | 14 | 12 | **16** | 12 | 12 | 11 | 12 | 12 | 16 |
| OSS | **13** | **13** | **13** | **13** | **13** | **13** | 12 | **13** | 13 |
| PNA | 18 | 6 | 17 | **19** | 0 | **19** | 18 | 18 | 19 |
| Set | 9 | 9 | 9 | 8 | **10** | 8 | 8 | 8 | 10 |
| Sna | 12 | 11 | **14** | 11 | 13 | 4 | 11 | **14** | 14 |
| Spi | 11 | 11 | 11 | 12 | **15** | 6 | 11 | 12 | 15 |
| Ter | 15 | 12 | 16 | 16 | 14 | **18** | 16 | 16 | 18 |
| Bar14 | 3 | 3 | 4 | 3 | 3 | **6** | 3 | 3 | 6 |
| Cave14 | **7** | **7** | **7** | 6 | **7** | **7** | **7** | **7** | 7 |
| Child14 | 0 | 0 | 0 | **5** | 0 | 4 | 0 | 1 | 5 |
| City14 | 10 | 10 | 10 | 11 | 14 | **18** | 10 | 13 | 18 |
| Fl14 | **20** | **20** | **20** | **20** | 8 | **20** | 14 | **20** | 20 |
| GED | **20** | **20** | **20** | **20** | **20** | **20** | **20** | **20** | 20 |
| Hike | 17 | 17 | 16 | 12 | 10 | 10 | 10 | **19** | 19 |
| Mai | **5** | **5** | **5** | **5** | **5** | **5** | **5** | **5** | 5 |
| OS14 | 8 | 3 | 8 | 5 | 2 | 8 | 5 | **13** | 13 |
| Pa | 4 | 4 | 4 | 3 | **6** | 2 | 3 | 4 | 6 |
| Tet14 | 11 | 11 | 12 | **14** | 13 | 10 | 11 | 13 | 14 |
| TB14 | **13** | 11 | 12 | 5 | 7 | 3 | 7 | **13** | 13 |
| Tr14 | 9 | 9 | 9 | 9 | **10** | 9 | 9 | 9 | 10 |
| Va14 | 13 | 14 | 10 | 14 | 13 | 7 | 7 | **15** | 15 |
| Bar11 | 7 | 8 | 8 | 8 | 7 | **9** | 8 | 8 | 9 |
| Elev | 19 | 19 | 19 | 19 | 19 | **20** | 19 | 19 | 20 |
| Floor | **12** | **12** | **12** | **12** | 6 | **12** | **12** | **12** | 12 |
| Mys | **20** | **20** | **20** | 14 | 14 | 11 | 14 | 14 | 20 |
| OS | 19 | 16 | 16 | 13 | 14 | 14 | 18 | **20** | 20 |
| PP | 16 | 16 | 16 | 18 | **20** | 14 | 18 | 18 | 20 |
| Pa | 1 | 1 | 1 | 4 | **7** | 1 | 1 | 1 | 7 |
| Peg | **20** | **20** | **20** | 16 | 17 | 17 | 16 | 19 | 20 |
| Scan | 9 | 9 | 9 | 8 | **12** | 8 | 7 | 9 | 12 |
| Sok | **20** | **20** | **20** | **20** | **20** | **20** | **20** | **20** | 20 |
| TB | **17** | 15 | 15 | 12 | 13 | 9 | 13 | **17** | 17 |
| Tr | 11 | 11 | 11 | **13** | 13 | 10 | 10 | 11 | 13 |
| Vis | 15 | 16 | 16 | 14 | 8 | 9 | 10 | **17** | 17 |
| Wo | 18 | 13 | 13 | 19 | **20** | **20** | 19 | 19 | 20 |

Table 3: Complete coverage (total number of problems solved) on all of the domains from the previous 3 IPC, cost-optimal track (2011, 2014 and 2018). Domain names have been abbreviated. The planners tested are: three versions of GreedyPDB (one only using Bin Packing, one only using Partial Gamer, and one with both approaches combined); BP-PDB planner; Scorpion and both versions of Complementary planners from IPC 2018; SymBiDir (benchmark planner from IPC 2018).

the others. Scorpion is the overall best in term of instances solved (1197) being by far the best on the older benchmarks. Complementary2 solves a close number of instances 1169, with GreedyPDB close behind with 1140 instances solved.

The reason for the swing in problems solved pre-2011 in favour of the approach Scorpion implements is due to the nature of the domains from that time, most of them catering towards explicit planning. It is also noteworthy that most domains in 2011-2018 benchmarks have 20 instances, while the pre-2011 are on average of 35, with some getting to 202.

By normalizing per domain, we get a slightly different picture, seen in Table 4. As there are some repeating domains in the benchmark sets from different IPCs, we insist on showing the results split over different IPCs, which are meant to encourage domain-independent planning.

On the 2018 benchmark, likely the most challenging one featuring a wide range of expressive application domain models, GreedyPDB would have actually won the competition (Table 1). This indicates that for several planning problems, the best option is to keep growing one PDB with the

| | Problems Solved | Coverage | Normalized Coverage |
|---|---|---|---|
| GreedyPDBs | 1140 | 55.04% | 61.08% |
| Planning-PDBs | 1122 | 54.17% | 59.42% |
| Scorpion | **1197** | **57.79%** | 59.26% |
| Sym-BiDir | 1053 | 50.84% | 55.46% |
| Complementary1 | 1099 | 53.06% | 57.60% |
| Complementary2 | 1164 | 56.15% | **62.08%** |

Table 4: Results as number of problems solved, coverage and normalized coverage.

greedy pattern selector, and compare and merge the results with a PDB collection based on bin packing [3].

## 7 Related Work

Pattern Databases have become very popular since the 2018 International Planning Competition showed that top five planners employed the heuristic in their solver. However, the topic has been vastly researched prior to this competition, a lot of work going in the automated creation of a PDB, with the best know being the iPDB of Haslum et al., (2007) and the GA-PDB by Edelkamp (2006). The first performs a hill-climbing search in the space of possible pattern collections, while the other employs a bin-packing algorithm to create initial collections, that will be used as an initial population for a genetic algorithm. iPDB evaluates the patterns by selecting the one with the higher *h*-value in a selected sample set of states, while the GA of the GA-PDB uses the average heuristic value as its fitness function.

Another two approaches related to our work is Gamer (Kissmann and Edelkamp 2011) and CPC (Franco et al. 2017). The first is in the search of only one best PDB, starting with all the goal variables, and adding the one that it will increase the average heuristic value. CPC is a *revolution* of the GA-PDB approach, aiming to create pattern collections with PDBs that are complementary to eachother. It also employes a GA and its evaluation is based on Stratified Sampling.

## 8 Conclusion and Discussion

The 2018 International Planning Competition in cost-optimal planning revealed that symbolic PDB planning probably is the best non-portfolio approach. In fact, five of the top six IPC planners were based on heuristic search with PDB and/or symbolic search, while the winning portfolio used such a planner (namely SymBA*, the winner of IPC 2014) for more than half of its successful runs.

In this paper, we present two methods building on top of the CPC approach by Franco et al., (2017), one incremental on an existing work (Planning-PDB), and one that is a reformulation of how it creates complementary pattern collections (GreedyPDB), by combining it with an adapted version of the Gamer approach (Kissmann and Edelkamp

---

[3]We include all the results of our experiments IPC11-18 in Table 3. The rest are available online.

2011). In both we have only one bin-packing solver, removing the multi-armed bandit algorithm to select its packing algorithm. In GreedyPDB, we also removed the optimization done with a GA over the pattern collections, seeing that bin-packing and partial-gamer complement already very well each other. Overall, the structure of GreedyPDB in comparison with CPC is very much simplified, with a small loss of coverage on the problem set of the IPC 2014.

Using different pattern generators to complement the two seeding heuristics was extremely successful. It improved our overall results for all the methods we tested compared to simply using the seeding heuristics. One of the best performing method is the combination of an incremental pattern selection with advanced bin packing. When combining both pattern selection methods, the results are greatly improved, and GreedyPDB would have won the last IPC even ahead of the best portfolio planners (solving 5 more problems), thus contributing a new state-of-the-art in cost-optimal planning.

It is probable that using SCP instead of canonical would improve results. It is also likely that if we used SCP online, i.e., for evaluating whether to add a PDB to the current selected set, instead of the current 0/1 approach a PDB is evaluated, would significantly reduce the total number of patterns we can try given the IPC time limit. How to navigate the trade-off between SCP's better heuristic values vs 0/1's faster computational time is future research.

However, as seen with the impressive results of Complementary2 in the 2011 and 2014 competition benchmark, there is no free lunch. Which pattern generator method is best depends on the benchmark domain it is applied to. By the obtained diversity in the individual solutions, an oracle deciding which pattern selector to take would have solved more problems, so that a portfolio planner could exploit this.

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
