# OpenReview forum: "Simplifying Automated Pattern Selection for Planning with Symbolic Pattern Databases"
_icaps-conference.org/ICAPS/2019/Workshop/HSDIP_

### Official Review · AnonReviewer2 · 2019-04-01
**The paper evaluates two methods for generating patterns for symbolic pattern databases but omits many crucial details and comparisons.**

**Rating:** 4
**Confidence:** 4

**Review:**

The paper certainly addresses an interesting research question and I think trying to understand what made the Complementary planners work well in IPC 2018 is a worthwhile endeavor. Unfortunately, I don't think the paper in its current form takes us closer to this goal. The paper presents two algorithms but many details are left out. This makes it impossible to understand how the algorithms work and how they differ from previous work. Furthermore, the experiment section is missing many essential comparisons, making it impossible to judge how the algorithms compare to previous work.

Introduction:
* Each concrete plan has an abstract counterpart in an induced abstraction of the same length, not "a shorter one".
* The meaning of a pattern in the sliding-tile puzzle depends on the representation. It can be a set of tiles or a set of positions.

Background:
* No need to introduce STRIPS, PDDL and PSVN. It doesn't seem you're using these later in the paper.
* Definition 1: there's no definition for \phi, \mathcal{A}, and the concept of plans.
* "The set of reachable states is generated on-the-fly" -> it is defined declaratively
* You describe "spurious states" in the background section, but never evaluate their impact or describe how you deal with them.
* I think "unreachable states" would be a better name than "spurious states" if that is indeed what you mean.

Pattern Databases:
* The codomain of heuristics should include \infty.
* "the sum of heuristic values obtained via projection to a disjoint variable set is admissible"
  -> This is only true if no two operators affect the same variable.
* "The projection of state variables induces a projection of operators and requires 0/1 cost partitioning"
  -> No, any cost partitioning algorithm can be used. Or you can maximize over the estimates instead of summing them.
* Definition 5: "maximize the average heuristic [value]"
  -> why not the heuristic value of the initial state or a set of samples? I think this needs some motivation.

Pattern Selection and Cost Partitioning:
* "SCP relies on only using those costs which each heuristic uses to create an abstract plan."
  -> This sounds misleading: SCP preserves *all* heuristic estimates under the given cost function.
* "[SCP] is time consuming compared to other cost partitioning methods (Pommerening, Helmert, and Bonet 2017)"
  -> the reference does not mention SCP. Seipp, Keller, Helmert (ICAPS 2017) state
  "[SCP] can be computed at negligible overhead during the construction of the heuristic"

Greedy Selection:
* The way I understand it, Gamer uses *steepest-ascent* hill climbing and Gamer-style uses *simple* hill climbing. If that's the case, I think using these terms might clarify the presentation.

BP-PDB:
* "Even though reducing the number of PDBs used to group all possible variables does not guarantee a better PDB, by having a smaller PDB collections, it is less likely to miss interactions between variables due to them being placed on different PDBs."
  -> This suggests trying to base the pattern generation on the variable interactions and not on their domain sizes as bin packing does.
* "the method will start generating pattern collections stochastically"
  -> How?
* "We then decide whether to add a pattern collection to the list of selected patterns if it is estimated that adding such PDB will speed up search."
  -> How?
* "To evaluate the fitness function"
  -> Which fitness function?

GreedyPDB:
* The limits of 50s and 75s for bin packing seem very large, given that most tasks have less than 1000 variables and the greedy bin packing algorithms FFI and FFD should run very fast.
* The first sentence of the last paragraph in this section appears twice in the paper.
* I guess some variable names are wrong in Algorithm 1?: SelPDBs vs. P_sel and SizeLim vs. Size
* Why do you iteratively increase the size limit in the PACKER function? I thought either the variables fit or they don't.
* How does Generate_P work?
* How exactly does GreedyPDB differ from Complementary 1 and 2?

Experiments:
* Figure 2 should probably use a histogram or similar instead of a line plot.
* Which time and memory limits do you use?
* Tables 1 and 2 are superfluous since they are subsumed in Table 3.
* I don't think there's an advantage in knowing the coverage score of an oracle planner.
* What is "advanced" bin packing?
* The paper states that GreedyPDB is a "new state-of-the-art in cost-optimal planning". However, Table 3 shows that Complementary 2 solves more tasks than GreedyPDB in 14 domains, while the opposite is true in only 6 domains.
* How do the parameters of GreedyPDB influence the resulting heuristic?
* I think the experiments focus too much on comparing whole planners and too little on comparing pattern selection algorithms.
* Keeping all other parts of the planner the same, the paper should compare different approaches for selecting patterns. Table 3 goes into the right direction by evaluating the three Greedy PDB variants, but this comparison only includes new pattern selection algorithms. The comparison should also include the following pattern selection algorithms:
  * the algorithms used in BP-PDB and Complementary 1 and 2
  * the Gamer pattern selection (without it, we cannot judge whether "Gamer-style" has any merit)
  * the hill climbing pattern generator (Haslum et al., AAAI 2007)
  * the systematic pattern generation methods of patterns up to size 2 and 3 (Pommerening et al., IJCAI 2013), which have been shown to yield very strong heuristics.

References:
* Some of the references for Stefan Edelkamp are missing a venue.

Style:
* Horizontal and vertical lines in tables only help if not all cells are fully surrounded by lines. One line below the header should be sufficient.
* Your inline citation macro seems to put extra whitespace between the authors and the year.
* No need to abbreviate domain names in Table 3, there's enough space.
* The paper has numerous typos. I recommend having it proof-read.

Reproducibility:
* BP-PDB only has a very high-level description and the pseudo-code for GreedyPDB is also missing some crucial details (e.g., EM algorithm). The paper does not promise to make any code or results available. Therefore, it won't be possible to replicate the results or build on them in future work.

---

### Official Review · AnonReviewer1 · 2019-04-05
**Interesting topic, but presentation is poor. Incremental work where exact contribution is not very clear.**

**Rating:** 6
**Confidence:** 4

**Review:**

The presented paper introduces an extension to existing pattern
generation algorithms for PDB heuristics. The work is fairly
incremental, mostly turning screws in a large space of existing
methods and combining them in new ways. Also, it does not become very
clear what the concrete contribution of this paper is. How does it
differ from existing methods, such as those presented in "On Creating
Complementary Pattern Databases" by Franco et al. (2017)? I guess a
separate related work section could help a lot in clarifying the
difference to existing methods and the contribution of this work.

The way the paper is currently written, by interleaving background on
existing methods (which btw. makes more than half of the paper), with
remarks about which ingredients of these methods are used in the
presented approach with or without modifications, makes it hard to
follow. More generally, the write-up is not very good and could be
significantly improved.

Also, at least in the presentation, the suggested changes are not very
systematic, there is no consistent thread. Rather, I have the
impression that various existing methods are combined to somehow
achieve higher total coverage.

Still, the experimental evaluation is not very convincing. When
choosing the right benchmark set (IPC'18), the new method slightly
outperforms the shown existing methods (including Complementary2).
When adding the IPC'11+14 benchmarks, Complementary2 is slightly
ahead. What happens when using all IPC benchmarks ('98-18)?

Nevertheless, I think the topic of the paper is interesting for the
HSDIP audience and the presented ideas might foster interesting
discussions, since automatic pattern generation for PDB heuristics
seems to be a hot topic these day.

If the paper gets accepted, I urge the authors to work on the write-up
and introduce a separate related work section.



Minor comments:
- abstract: "...which*,* if expressed symbolically*,*..."
- next sentence: "in the automatic* generation"
- please remove the copyright statements from the first page.
- Def.1: s_o*and*s_*
- Def.1: the sets {\cal A} and {\cal A}^+ are not defined
- figure 1 floats into the margin
- "Franco et al... shows that ... (Franco etal)." remove one of the
cites.
- end of paragraph below Def.4: "symbolic symbolic"
- "contribute to more that on*ce*"
- "the correlation between the cross product [..] is rather weak."
I agree that it is weaker than for explicit representations, but still
"rather weak" is not a good description.
- the cite of FD "[Helmert, 2006]" has a different format than the
other cites
- the style of writing is sometimes very informal, e.g.
"Baeckstroem prefers the SAS+ formalism", or "in the most prestigious
and attended [..] track"
- for consistency, please put the caption of table 2 below the table
- the bibliography does not seem to be in AAAI style
- the references Edelkamp 2002a and b refer to the same paper, same
for Holte and Hernadvoelgyi.
- there is a typo in the reference to Preditis 1993 *heuristics*

---

### Author Response · Authors · 2019-04-09
**Thank you for your insightful comments**

We would first like to thank the reviewers for careful reading and very constructive comments on how to improve our paper. We would now want to respond to some of them.

Writing Style:
Thank you for the numerous comments to improve the paper in this area, some are really helpful and we have fixed most of the ones raised by both reviewers that could be changed in the short revision phase. Whether or not introducing planning formalisms is needed, we think it is helpful to explain the concept of abstraction that is basic to pattern database design. We realise that our presentation, having more elements introduced in each section, is different from most publications, and we will strive for writing more in that style in the future. For adding a Related Work section, we will do that, but we could not have done it well in this short time.

Contribution:
In our opinion, this paper brings forth a solid contribution. We present the currently best planner on the IPC-2018 benchmark. We simplified the Complementary planners significantly, in the area of Pattern Selection, and currently have better results than the portfolio planner Delfi by 5 problems (we should exceed even their AAAI-19 posterior IPC-18 results, where only relative numbers are given). The paper clearly illustrates that bin packing plays a crucial factor in PDB planner design and complements well with other methods of partial pattern databases, as seen in figure 2.

Experiments:
We consider that the complexity and range of problems considered in the 2018 edition of the IPC are already of very high quality (10 domains with each having 20 instances).  We consider the IPC editions to be tested on sets of domains that are testing for the domain-independence of each planner, not testing on specific types of problems. As such we considered testing on the domains of the last three competitions.

Since submission, we have run experiments on a subset of the domains from 1998-2009, with interesting results. We have seen that Scorpion planner outperforms all the planners on those benchmarks, becoming the best with 1197 problems solved on 69 domains (compared to 1169 and 1140 of Complementary2 and GreedyPDB). As the results of 98-09 are very different from the ones from 11-18 because (we identified) the domains have very different numbers of problems compared to the newer ones (up to 150). We normalized the results to see coverage per domain and the by averaging that results we see that, while Scorpion has a better average than before, it is still outperformed by Compl2 and GreedyPDB (maintaining the ideas put forward in the paper)

Normalized Average  - GreedyPDB: 61.1% Planning: 59.4% Compl1: 57.6% Compl2: 62.1% Scorpion: 60.1% SYM-BiDir: 55.5%

Reproducibility:
We have before, and we will publish the code of our planners, together with the logs of our experiments. Apologies for not stating this explicitly in the paper.

About the comment on abstract path length: if there are (self) loops of a mapped concrete path in abstract space they should be omitted and lead to shorter abstract paths.

Thank you again for your reviews of our paper. We will be working on improving the presentation of the paper.

---

> ### Comment · AnonReviewer2 · 2019-04-10
> **Rebuttal ignored most of my comments**
>
> I appreciate you revising the paper. However, a quick glance at the diff shows that you introduced at least two problematic changes:
>
> * You added that "the average heuristic value [...] is the best approximation up to this point." Which paper shows this? Haslum et al. (AAAI 2007) report that their counting approximation is consistently preferable to using the average heuristic value.
>
> * You dropped the "0/1" from the sentence describing that "projections need 0/1 cost partitioning" but then still describe zero-one cost partitioning.
>
> Your rebuttal ignored most of the points I raised in my review. In particular, you did not comment on my main criticisms: that it's unclear how the new methods differ from previous work and that the paper doesn't make the necessary experimental comparisons (see paragraphs BP-PDB, GreedyPDB and Experiments in my review). Until my questions and criticisms are properly addressed, I cannot vote to accept the paper.

---

> > ### Author Response · Authors · 2019-04-11
> > **Average Heuristic Value**
> >
> > Dear Reviewer!
> >
> > To have clear answers to
> > your questions will require years of exiting research to come. We are at the start not
> > at the end of an avenue.
> >
> > We showed that by the current state of publication we are still state-of-the-art in the
> > 2018 benchmark set. The complexity and range of problems considered in IPC 2018 already
> > is remarkable: 10 domains, 20 problems each, ask the organizers if you are afraid they
> > are easy.  It has been argued many times, that early benchmarks in IPC
> > like Logistics are computationally rather trivial. Of course, heuristics exploiting structure
> > can be good in there.
> >
> > This paper is on general-purpose planning, neither portfolio, nor fine-tuning to one benchmark
> > set. As said theree is no free lunch, but the 2018 results are remarkable. We inserted 2014 and 2011
> > results for convenience.
> >
> > Table 3 shows 8 different planners on more than 35 different domains. With simplyfying the planner, using only a fraction of the overhead of Complementary (such as no Bandit-based optimization).  We clearly showed the complementary effect of bin packing and
> > partial gamer in Figure2 and Column 2 and 3 of Table3.
> >
> > We also discussed the results on all pre2010 domains in the rebuttal and showed that
> > in a normalized comparison together with Complementary we are still best across all IPC domains.
> > There are over 2K planning task in all domains, Assuming 30min each than this is more than 80days
> > CPU run time for each planner configuration. We ran them all, but will you only accept work that
> > does this, then this only leaves a small number of researchers remaining to contribute having
> > access to a big cluster.
> >
> > Scientifically, it might be  impossible to get clarity on every aspect of a complex planning system with state-of-the-art performance. We might not like the suggested combination of BDDs, BinPacking, Partial PDBs etc, but they are doing the trick, better than othersuggestions of advanced heuristic designs.
> >
> > The stress of the paper is simplyfying. It is not the the planner with best maths
> > inside wins.
> >
> > We made the planner work in the FD framework for simplified use. Bin Packing is not the theoretically best selector for PDBs, but it is fast and surprisingly effective. PDBs are theoretically worse to Merge&Shrink, but in practice they perform better. Symbolic search are simplyfying backward traversal efficiencies, and can be combined with forward search. Symbolic Blind search outperforms almost all planners with refined heuristic, leaving only PDB planners and portfolios to pass the bar. We had so many explicit-state heuristics awarded in papers in top conferences and journals. They are not telling the entire story of competitive cost-optimal planning.
> >
> > This paper helps out in showing directions on future planner designs. Even with 0/1 and no saturated cost partitioning, one can get good results. Even without GAs optimization one can do well. This does not mean that aspects won't help.
> >
> > --
> >
> > The best average heuristic value certainly is not the best approximation for choosing if one
> > heuristic is better than another.  As you correctly indicated, the average heuristic value for a problem will include unreachable states (undetected dead ends by the heuristic) or states that are not visited while searching for a solution for a specific problem.  For the competition we actually also used a random walk (the Haslum method) to evaluate if heuristics are improved, so we are not throughout using the average heuristic value to decide between patterns.
> >
> > But it is also not the worst criterion. We believe in Occam Razor. The simpler method does not have to be worse.
> >
> > HEURISTIC
> >
> > As you asked for a reference. In Korf’s milestone first PDB paper
> >
> > https://www.aaai.org/Papers/AAAI/1997/AAAI97-109.pdf
> >
> > he states
> >
> > "More formally, let n be the number of states in the entire problem space, let b be the brute-force branching factor of the space, let d be the average optimal solution length for a random problem instance, let e be the expected value of the heuristic, let m be the amount of memory used, in terms of heuristic values stored, and let t be the running time of IDA*, in terms of nodes generated. The average optimal solution length d of a random instance, which is the depth to which IDA* must search, can be estimated as log_b n, or d ~ log_b n. As argued above, e ~ log_b m, and t ~ b^(d-e). Substituting the values for d and e into this formula gives
> >
> > t ~ b^(d-e) = b^(log_b n log _b m) = n/m."
> >
> > This was the first analysis for PDBs and was interpreted as a milestone, as previously they thought heuristic reduces width, not depth. The approximation in the Rubik’s cube were very precise. This result led to a paradigm shift that heuristics do not reduce the branching but the depth of the search.
> >
> > What we say that using average heuristic values is a plausible setting to explain one in a range of options, at least for the theoretical part. This makes the paper accessible.

---

> ### Comment · AnonReviewer1 · 2019-04-11
> **Thank you for the thorough response!**
>
> First of all, thank you for spending such an effort in a really detailed response.
>
> From my perspective, however, the discussion goes into the wrong direction, not talking about the important and interesting points.
>
> There is no doubt that your planner performs well on the IPC'18 domains, and your first comment shows that it's also comparable to Complementary on the entire benchmark set. My point regarding significance is rather that you put too much emphasis on this. Being strong in the IPC'18 domains is good, but if your approach does not capture the (as you say) "computationally rather trivial" domains from other benchmarks, which other approaches can solve, then there is at least clearly no dominance between your approach and these others. I'd be happy if you somewhat de-emphasise the role of your results on the IPC'18 suite, mostly in the conclusion.
>
> The main point, however, is clarity and delimination from related work. When processing your comments, I was also reading up on previous work (Santiago et al 2017, Complementary planner abstract), and I see that you do things different in GreedyPDB, but this has to be communicated. From only reading your paper, it's hard to impossible to tell the differences.
>
> Therefore, as already said, please introduce a related work section. In there, you could introduce different existing methods (e.g., the two mentioned above), and clarify the difference to GreedyPDB/BP-PDB. Additionally, I think it would help to only introduce the base techniques in the respective subsections, without talking too much about which of the existing methods uses some of these (with some specific modifications) in its approach.
>
> Some more specific comments:
> "The paper clearly illustrates that bin packing plays a crucial factor in PDB planner design and complements well with other methods of partial pattern databases, as seen in figure 2."
> => I see the exact opposite when looking at Figure 2 and Table 3. The PartialGamer patterns seem to contribute more than bin-packing.
>
> - I like the idea of showing the impact of the base techniques on overall performance, although I'm not convinced that grouping domains by IPC year as in Figure 2 is very helpful, I prefer a domain-by-domain comparison as in Table 3. I'd also appreciate a similar split for BP-PDB, showing performance when only using the perimeter.

---

> > ### Author Response · Authors · 2019-04-11
> > **We will donetone.**
> >
> > Thanks for the reply.
> >
> > We all know there are certainly planners that are better for Logistics or Blocksworld,
> > even if the latter in NP-hard for cost-optimal. In the conclusion, we explicitly said there is no free lunch in AI planning. Performance always will dependent on benchmarks chosen. That is why we have the IPC to test planners on unknown and
> > increasingly complex domains.
> >
> > Even across all known existing IPC domains, the performance for our simplified PDB planner architecture in this paper, however, is remarkable, and that is what domain-independent planning is about. With each domain contributing one vote and, together with complementary probably leading at least to our knowledge. (We have collected evidence that some other, already proposed bin packing scheme can actually beat both these planners on these statistics.)
> >
> > About Figure 2. The core aspect is to see is that the combination of the two is clearly better than each of the two alone, when given the same run-time and memory bounds. This clearly is not always the case. So it depends on how you see data. Table 3 assist Figure 2, we agree.
> >
> > Good suggestion to try another simplified setting along this study. We know that Perimeter PDB alone is not good enough, it correlate with Partial Gamer, but certainly we can try running perimeter only. To doube-check what you want, isn't one single perimeter PDB not similar to bidirectional blind? If there is only one PDB there is no packing needed. At least we would know that the results are not comparable to the top performers. In any case, doing more experiments takes at least some time...
> >
> > Thanks for reading the related papers. Very much appreciated. So far very little is published about Complementary and it has changed significantly since the IJCAI work.  It is rather complex system and explaining this system is hardly possible in this paper range. This paper starts to sort things out and shaves off fat that may not be needed.
> >
> > We will try improving the distinction between this planner contribution and complementary. Give us a few days, we might reintroduce an architectual figure that we kicked out. In general, however, is not as simple as it one might wish, as they were share a common base, were coded in common programming efforts,  for example new bin packing routines we are fighting for in this paper, found their way into complementary. One out of many smaller difference is that GreedyPDB does not use genetic algorithms (as PlanningPDB) nor bandits (as Complementary), and, thus, a smaller and more deterministic selection of possible PDB candidates.
> >
> > To this extend this paper is a first step, it speak for symbolic PDB planning in general, and presents one simplifying PDB approach in depth.
> >
> > To the amount of related work at least one side remark. We already have one full page of references, and look into saturated cost partitioning in depth.

---

> > > ### Comment · AnonReviewer1 · 2019-04-11
> > > **thanks**
> > >
> > > - "To doube-check what you want, isn't one single perimeter PDB not similar to bidirectional blind?"
> > > => I'm assuming that the perimeter (if it does not completely solve the task itself) is used as a partial PDB, where abstract states not seen in the perimeter search are mapped to the maximum distance. In this case, I'd assume that the performance is somewhat better than pure symbolic backwards search, but probably worse than the PartialGamer PDBs. In any case, I'd be happy to see the numbers in the final version, when accepted.
> > >
> > > I understand that with the complexity of the existing pattern generation methods, it's not easy to concisely describe them and at the same time introduce a new technique within the given page limit. And I definitely appreciate your attempt to simplify the pattern generation.

---

### Meta-Review · Program_Chairs · 2019-04-25

**Recommendation:** Accept
**Confidence:** 5

**Metareview:**

Dear Authors,
thank you very much for your submission. We are happy to inform you that
we have decided to accept it and we look forward to your talk in the workshop.
Please, go over the feedback in the reviews and correct or update your papers
in time for the camera ready date (May 24). In particular, please address the
comments raised by both reviewers regarding clarity, discussion of related work,
and additional experimental results, by making use of the additional space (9 pages)
allowed by HSDIP.
Best regards
HSDIP organizers